# The Role of Vitamin D Deficiency in Hepatic Encephalopathy: A Review of Pathophysiology, Clinical Outcomes, and Therapeutic Potential

**DOI:** 10.3390/nu16234007

**Published:** 2024-11-23

**Authors:** Coplen D. Johnson, Christopher M. Stevens, Matthew R. Bennett, Adam B. Litch, Eugenie M. Rodrigue, Maria D. Quintanilla, Eric Wallace, Massoud Allahyari

**Affiliations:** 1School of Medicine, Louisiana State University Health Shreveport, 1501 Kings Highway, Shreveport, LA 71103, USA; cdj002@lsuhs.edu (C.D.J.); mrb003@lsuhs.edu (M.R.B.); abl003@lsuhs.edu (A.B.L.); emr001@lsuhs.edu (E.M.R.); mdq001@lsuhs.edu (M.D.Q.); 2Department of Radiology, Louisiana State University Health Sciences Center-Shreveport, 1501 Kings Highway, Shreveport, LA 71103, USA; eric.wallace@lsuhs.edu (E.W.); massoud.allahyari@lsuhs.edu (M.A.)

**Keywords:** hepatic encephalopathy, vitamin D deficiency, cirrhosis, oxidative stress, vitamin D supplementation, transjugular intrahepatic portosystemic shunt

## Abstract

Hepatic encephalopathy (HE) is a neuropsychiatric condition frequently associated with cirrhosis and portosystemic shunting (PSS). It imposes a significant clinical and economic burden, with increasing attention toward identifying modifiable factors that could improve outcomes. Emerging evidence suggests that vitamin D deficiency (VDD), prevalent in patients with cirrhosis, may contribute to the development and severity of HE. This review explores the association between VDD and HE by analyzing the underlying pathophysiology, including oxidative stress, ammonia accumulation, and impaired hepatic function. Additionally, we summarize recent studies highlighting the correlation between low serum 25-hydroxy vitamin D (25-OHD) levels and worsening grades of HE. Despite strong observational data, interventional studies on vitamin D (VD) supplementation for HE remains limited. Current evidence suggests that VD’s antioxidant properties may alleviate oxidative stress in HE, with potential benefits in mitigating disease severity. Future research should focus on longitudinal studies and randomized controlled trials to evaluate the clinical impact of VD supplementation on HE outcomes and explore VD’s role in patients undergoing transjugular intrahepatic portosystemic shunt (TIPS) procedures. Understanding the therapeutic potential of VD could lead to improved management strategies for HE and cirrhotic patients at large.

## 1. Introduction

Hepatic encephalopathy (HE) is a complex neuropsychiatric brain dysfunction that manifests as altered mental status, cognitive impairment, and other neurological symptoms [1]. HE is a common condition, with between 7 and 11 million cases in the United States and approximately 150,000 newly diagnosed cases each year [2]. Around 30–45% of patients with cirrhosis and 24–53% of patients with portosystemic shunt procedures develop HE [3]. HE is associated with increased economic and utilization burden to patients and caregivers and a diminished quality of life [4]; therefore, developing novel therapeutic approaches to help diminish the incidence of HE and its devastating complications is of great importance. Recently, vitamin D (VD) status has become a popular topic in the realm of HE, as multiple studies have reported an association between VDD and HE [5,6,7]; herein, we present a literature review examining the association between vitamin D deficiency (VDD) and HE and discuss the possible beneficial effect VD supplementation can have in this patient population.

## 2. History, Etiology, Clinical Features, and Diagnosis of Hepatic Encephalopathy

HE is the most common complication arising from cirrhosis or portosystemic shunting (PSS) [2]. HE can develop from both chronic and acute liver disease. Acute causes include hepatotoxins such as acetaminophen and ischemic liver injury. Chronic liver failure may result from conditions like alcoholic cirrhosis, chronic viral hepatitis infections, and non-alcoholic fatty liver disease [2]. This list is not exhaustive though, as any cause of hepatic dysfunction has the potential to lead to HE.

HE is characterized by neuropsychiatric changes, such as asterixis, in patients with liver disease due to hyperammonemia, toxic metabolites, and brain swelling [8,9,10]. Patients may present with subtle neuropsychological impairments, such as attention deficits and slowed psychomotor speed, progressing to more overt symptoms like confusion, somnolence, and disorientation. In severe cases, HE may progress to stupor or coma. Psychometric testing often reveals deficits in visuospatial ability, working memory, and executive function which may not be evident on routine neurological examination [8,10].

Neuromuscular manifestations of HE are also significant and may include tremors, rigidity, and reduced motor coordination. Asterixis, characterized by sudden, brief, nonrhythmic lapses in posture, is a hallmark sign of HE. In advanced stages, extrapyramidal symptoms such as bradykinesia and hypertonia may develop, with some patients showing signs of corticospinal tract involvement, such as spasticity or hyperreflexia [8]. These symptoms are often accompanied by alterations in the sleep–wake cycle and motor slowing which can significantly impact daily functioning [9,10].

The pathophysiology underlying these manifestations is complex, involving a combination of hyperammonemia, neuroinflammation, and astrocyte swelling. Hyperammonemia, resulting from impaired ammonia metabolism due to liver dysfunction or portosystemic shunting, leads to the accumulation of glutamine within astrocytes, contributing to cerebral edema and neurotransmitter imbalances [9,10]. This process disrupts synaptic transmission and contributes to both cognitive and motor impairments. Neuroinflammation, characterized by microglial activation and increased proinflammatory cytokines, further exacerbates these effects, leading to increased glutamatergic and GABAergic signaling that contributes to the characteristic neurological symptoms of HE [10].

Elevated ammonia levels in arterial and venous blood can suggest HE but are not diagnostic. Hepatic function tests and serum electrolyte levels are useful for assessing the severity of underlying liver disease but are not definitive for diagnosing HE. While computed tomography (CT) and magnetic resonance imaging (MRI) imaging cannot diagnose HE, they are valuable for ruling out other causes of encephalopathy, such as intracranial hemorrhage and lesions [2]. Measurement of serum 3-nitrotyrosine can aid in diagnosing HE, as a cutoff of 14 nM demonstrates 93% sensitivity and 89% specificity for detecting HE [11]. Psychometric tests have shown higher sensitivity compared to electroencephalograms (EEG) in demonstrating mental function impairment in cases of mild HE; however, these tests are complex and time-consuming, thus posing challenges for clinical implementation [2].

## 3. Pathophysiology of Cirrhosis and Hepatic Encephalopathy and the Role of Oxidative Stress

Hepatic cirrhosis is characterized by fibrosis secondary to chronic liver injury from certain diseased states, including viral infections, toxins, hereditary conditions, and autoimmune processes, that lead to formation of scar tissue in the liver. Initially, the liver may retain its function, but if long-standing injury occurs, fibrosis may outpace the liver’s regenerative capabilities, leading to loss of function and cirrhosis development [12].

At the cellular level, cirrhosis is mediated by the death of hepatocytes, which occurs either through apoptosis or necrosis. The type of liver insult dictates which death routine is dominant. For example, acetaminophen-induced hepatotoxicity results in hepatocyte death through mitochondrial damage, leading to excessive reactive oxygen species that can increase mitochondrial permeability and transition pore necrosis. Contrarily, alcoholic liver disease results in the formation of toxic metabolites, which promote pro-apoptotic factors [13]. The cumulative death of hepatocytes and inflammation lead to fibrogenesis as the liver attempts to repair itself. Activated myofibroblasts secrete extracellular matrix (ECM) proteins, such as collagen, to replace damaged tissue. However, this process can result in excessive deposition of ECM, giving rise to widespread hepatic fibrosis and eventually cirrhosis [14,15]. Additionally, VDD leads to uptake of vitamin D-binding protein by hepatic stellate cells. This triggers their transformation into myofibroblasts resulting in the development of hepatic cirrhosis [16]. These pathological changes not only contribute to cirrhosis but may also play a role in the progression of HE by altering hepatic architecture and function.

HE is a neuropsychiatric syndrome that results from liver dysfunction and PSS [2]. PSS occurs due to portal hypertension (PH) characterized by increased pressure within the portal venous system due to pathologies such as liver disease or portal vein thrombosis. This results in the creation of numerous spontaneous portal-systemic shunts that divert blood away from the liver and into systemic circulation [17]. The combination of cirrhosis and PSS results in dysfunction of the urea cycle and hepatic detoxification, respectively. As the catabolism of amino acids occurs in muscle, alanine, an ammonia transporter, enters systemic circulation en route to hepatic arteries and the liver. Due to compromised function from hepatocyte injury, the liver cannot convert ammonia into urea that can be renally excreted. Ammonia then enters the portal vein, bypassing the liver in PSS. This surplus of systemic ammonia can diffuse across the blood–brain barrier and cause a toxic effect within the brain parenchyma [10]. Specifically, astrocytes can detoxify low amounts of ammonia through glutaminase enzymes that can form glutamine from ammonia and glutamate. However, in the presence of high concentrations of ammonia, astrocyte function can become dysregulated and result in cell swelling, leading to increased intracranial pressure, inflammation, disruption of oxidative stress homeostasis, abnormal mitochondrial permeability, and alteration of pH [18]. Other theories that try to expand upon the pathophysiology of HE exist; however, ammonia toxicity is the most recognized and what current treatment regimens target.

Oxidative stress is a key factor in the pathophysiology of HE. The accumulation of ammonia triggers oxidative stress by disrupting the balance between reactive oxygen and nitrogen species (ROS and RNS) production and the body’s antioxidants. Typically, ROS and RNS are normal components of cellular signaling, but they can lead to neurotoxicity and neurodegeneration when in excess. The elicitation of ROS and RNS by ammonia alters the glycine/glutamate cycle and affects brain function due to ammonia’s role in glutamine production and its ability to cross the blood–brain barrier. Given GABA’s role as the principal inhibitory neurotransmitter, its modulation by oxidative stress further complicates the neurological manifestations of HE. Local high ammonia concentrations directly activate GABAergic neurons in the substantia nigra pars reticulata leading to motor deficits seen in HE [19]. Furthermore, the regulation of antioxidant levels in cells also plays an important role in the moderation of oxidative stress in HE. Albumin serves as an antioxidant in plasma, accounting for more than 70% of the serum’s antioxidant activity. In chronic liver disease, albumin level is decreased, resulting in a decrease in its antioxidant activity [20]. The introduction of excess ROS and RNS secondary to hyperammonemia is also linked to neuroinflammation through systemic cytokine production. These cytokines induce an inflammatory cascade that worsens the effects of oxidative stress. In brief, the generation and downstream effect of oxidative stress in HE is multifactorial. The increase in ammonia due to deficient liver clearance is an important factor in the pathogenesis of HE due to its inflammatory, neurotoxic, and direct effects on oxidative stress.

## 4. Vitamin D and Its Association with Cirrhosis and Hepatic Encephalopathy

### 4.1. Brief Overview of Vitamin D Metabolism

VD is a fat-soluble vitamin that acts as a micronutrient and prohormone capable of transforming into active metabolites [21]. The two inactive forms of VD, cholecalciferol (D3) and ergocalciferol (D2), are obtained from dermal synthesis and dietary intake, mostly from sun exposure and subsequent dermal synthesis [22]. Both D2 and D3 undergo a two-step hydroxylation process that converts the inert form first into the circulating form of vitamin D ([25(OH)D]) in the liver, and subsequently into the active form of 1,25-dihydroxyvitamin D, or calcitriol, in the kidneys (Figure 1) [23].

### 4.2. Supplemental Forms of Vitamin D and the Definition of Deficiency

VD exists in various forms including vitamin D3, vitamin D2, 1,25-dihydroxyvitamin D, and 25-OHD. Serum 25-OHD is most commonly the marker that is used to assess VD status in an individual, as it accurately reflects the levels of VD metabolites present in the body. VDD is defined at levels less than 50 nmol/L, while insufficiency is categorized at levels between 50 and 75 nmol/L. VD sufficiency is measured at levels between 75 and 125 nmol/L [24]. The risk of osteomalacia and rickets is greatly increased at levels of less than 25 nmol/L, marking the cutoff for severe VDD [25]. In a healthy individual, a cutoff level of 50 nmol/L is recommended by the Institute of Medicine [26].

The most common and readily available forms of VD supplementation include vitamin D2, vitamin D3, and more recently, calcifediol, a vitamin D3 metabolite. Both vitamin D2 and vitamin D3 supplements, when administered at similar doses, are effective at maintaining serum 25(OH)D levels; however, recent studies suggest that vitamin D3 is more effective at correcting VDD [27]. Furthermore, at a pharmacokinetic level, calcifediol is preferable in certain clinical situations. Both vitamin D2 and D3 must undergo a two-step enzymatic process to become 1,25 (OH)_2_D, the biologically active form of VD. Given that calcifediol is one step closer to biologically active VD in the metabolic pathway, it can increase serum 25(OH)D levels at a faster rate than vitamin D3 [28]. This is beneficial in emergent situations that require immediate 25(OH)D concentration increases. 25(OH)D (calcifediol) also does not require hepatic metabolism, making it more suitable for patients with diminished liver function. Because the intestinal absorption of vitamin D3 requires bile acids and micelle formation, it is incompatible for use in patients with fat malabsorption disorders [29]. Calcifediol has also shown to be advantageous in obese patients due to its hydrophilic properties [30].

### 4.3. Vitamin D Levels in Association with Cirrhosis

Approximately 93% of cirrhosis patients exhibit VDD, with severe deficiency observed in as many as one-third of these patients [31]. The etiology of VDD in this population is likely multifactorial. In individuals with cirrhosis, the hepatic conversion of VD to 25-hydroxyvitamin D ([25(OH)D]) is diminished, particularly in advanced stages of the disease when minimal functional hepatic tissue remains [12]. Additionally, there is evidence to suggest that hyperparathyroidism in cirrhotic patients may exacerbate VDD. Clements et al. concluded that when production of 1,25-dihydroxyvitamin D is stimulated by parathyroid hormone, this further exacerbates VDD via hepatic inactivation of 25(OH)D [32]. This process underscores the interplay between hormonal regulation and hepatic function in VD metabolism. Furthermore, cholestatic liver disease, which can develop in cirrhotic patients, leads to decreased absorption of fats in the gastrointestinal tract, subsequently impairing the absorption of fat-soluble vitamins such as VD [33]. The extent of fibrosis and risk of mortality due to cirrhosis are significantly associated with the degree of VDD [34,35]. Additionally, VDD in cirrhotic patients is linked to an increased risk of infections [36]. Despite extensive studies characterizing the correlation between VDD and adverse outcomes in cirrhosis patients, there is a lack of evidence demonstrating a causative relationship; however, the supplementation of VD in cirrhotic patients has been shown to improve clinical outcomes [37].

### 4.4. Vitamin D Levels in Association with Hepatic Encephalopathy

Over the last fifteen years, numerous studies have shown an association between VDD and HE (Table 1). One cross-sectional study compared 25-OHD levels in patients with different grades of HE. The findings showed average serum 25-OHD levels of 24.11, 13.61, 8.41, and 8.00 ng/mL in Grades 1, 2, 3, and 4 HE, respectively [7]. Another cross-sectional study found that patients with HE had significantly lower serum 25-OHD levels compared to a control group, and a significant negative correlation was observed between 25-OHD levels and worsening grades of HE [38]. In a study involving 70 cirrhotic patients (35 with HE and 35 without), HE patients had significantly lower mean serum 25-OHD levels compared to those without HE. Notably, 91% of HE patients had moderate to severe VDD, compared to 51% in the control group [5]. Another study examined the relationship between low serum 25-OHD levels and HE in patients with Hepatitis C virus (HCV)-related liver cirrhosis. Lower 25-OHD levels were associated with a higher incidence of HE. Additionally, VD levels were significantly higher in patients who improved from HE compared to those who died from HE. This suggests that worsening VDD is linked to increased severity of liver cirrhosis, indicating that VD levels could serve as a prognostic factor for liver cirrhosis severity and mortality in HE patients [6]. There is a lack of randomized controlled trials (RCTs) analyzing the effects of VD supplementation on HE; however, one prospective study examined VD replenishment in patients with decompensated liver cirrhosis using the Child–Pugh (CP) scale, which assesses cirrhosis mortality and includes HE as one of the grading criteria. Patients with elevated CP scores showed a decline in scores over a 6-month period while receiving vitamin D supplementation. The VD dosage for this study included a loading dose of 300,000 IU intramuscular cholecalciferol followed by 800 IU/day oral maintenance, along with 1000 mg of calcium daily. This regimen indicated an overall positive effect of VD supplementation on cirrhosis mortality [37]. In another study, acute decompensation of liver cirrhosis was defined as the presence of HE, gastrointestinal hemorrhage, bacterial infection, or ascites. The study found an association between VDD and an increased risk of hepatic decompensation during follow-up. However, this risk was not present in the group receiving VD supplementation. This suggests a positive effect of VD supplementation on hepatic decompensation, though the study did not specifically analyze HE [39].

### 4.5. Vitamin D Levels in Post-Operative Transjugular Intrahepatic Portosystemic Shunt (TIPS)

TIPS is a surgical procedure performed in cirrhotic patients to manage complications arising from PH. The procedure involves guiding a transjugular catheter into the liver to establish a direct connection between the portal vein and hepatic vein. This connection, functioning as a shunt, redirects blood flow from the portal system to the systemic circulation, alleviating congestion caused by elevated portal pressures in patients with PH; however, a notable drawback with TIPS is that it can induce or exacerbate HE by bypassing the liver’s detoxification function, with postoperative incidence rates of HE ranging from 30% to 50% [47]. In addition to a decreased ability by the liver to convert VD to its circulating form, decompensated cirrhotic patients and patients with HE may already have an underlying nutritional deficiency and a lower capability of absorbing fat-soluble vitamins [48]. While there is no known direct causative association between TIPS and VD levels, the context of the procedure in the management of PH may influence VD levels and overall nutrition in patients undergoing the procedure.

### 4.6. Vitamin D Has Been Shown to Suppress Oxidative Stress

VD is recognized primarily for its role in calcium metabolism and bone health, but it also possesses antioxidant features. The antioxidant role of VD has been studied in several different diseased states, including diabetes, cardiovascular disease (CVD), Alzheimer’s disease (AD), and COVID-19. Supplementation with VD has been shown to alleviate oxidative stress by upregulating the production of crucial antioxidant enzymes, including glutathione (GSH) reductase, GSH peroxidase, superoxide dismutase, and catalase. The activation of VD receptors by VD may trigger the upregulation of genes encoding enzymes involved in GSH synthesis, such as glutamate-cysteine ligase. Some clinical trials have shown that monotherapy with VD decreases oxidative stress levels and improves cognitive impairment in patients with AD [49]. In diabetics, VD supplementation greatly reduces malondialdehyde levels, a marker of lipid peroxidation and oxidative stress, through the upregulation of GSH reductase [50]. In other studies, VD supplementation has shown an ability to decrease inflammatory cytokines involved in immune response and free-radical generation [51]. The multifaceted role of VD extends beyond its functions in calcium metabolism and bone health, encompassing potent antioxidant properties with significant implications across various disease states. Its ability to modulate oxidative stress and inflammatory pathways underscores its potential as a therapeutic target for patients with HE since these patients often have high levels of oxidative stress, as stated prior.

## 5. Vitamin D Levels in Association with Survival

Due to this vast range of regulatory functions, VDD has been associated with increased mortality and risk of developing immune and cardiovascular diseases. However, VD levels as a cause of increased mortality compared to a predictive marker of poor health is not well distinguished. While there is still debate, many believe serum 25-OHD levels should be maintained at 30 ng/mL and preferably at 40–60 ng/mL to reach the optimal overall health benefits of VD [52]. Regarding all-cause and case-specific mortality, multivariate analyses revealed a nonlinear inverse association between mortality and serum 25-OHD levels in a prospective study consisting of 365,530 patients. Patients with >24 ng/mL had a significantly 17% lower risk of all-cause mortality and a 23% lower risk of CVD mortality, while patients with serum 25-OHD above 18 ng/mL had an 11% lower risk of cancer mortality. Another study considering associations between VD levels and mortality risk on 1915 males showed the mortality risk was twice as high for those with serum calcifediol levels below 20 ng/mL compared to those with better VD status. However, no reduction in all-cause mortality was observed with monthly VD supplementation [53]. Another multivariate logistic regression analysis conducted on 1705 patients found that patients with 25-(OH)-D3 levels < 12.5 ng/mL had a higher mortality risk than those with levels > 12.5 ng/mL [54].

These studies all emphasize the association of lower VD levels with increased risk of all-cause mortality. However, there is no significant evidence showing that VD supplementation improves patients’ mortality risk. Additional studies will need to be conducted to determine what role, if any, VD supplementation plays in improving patient conditions and mortality risk.

## 6. What Should Future Studies Explore?

Future research on VD should aim to reveal broader implications beyond its traditional roles. First, longitudinal studies are necessary to determine the impact of VD levels on overall mortality and to explore the potential benefits of supplementation in reducing mortality rates among diverse populations. While correlations between serum VD levels and mortality have been noted, a closer examination is needed to identify the specific roles of VD in the survivability of patients across various disease states, including HE. For example, Jha et al. demonstrated that a regimen of vitamin D supplementation—comprising a single loading dose of 300,000 IU intramuscular cholecalciferol followed by daily oral maintenance of 800 IU, alongside 1000 mg of calcium—yielded promising results in improving clinical outcomes in patients with decompensated cirrhosis. However, the study’s findings on survival rates did not reach statistical significance, highlighting the need for larger, multicenter trials to confirm these effects and optimize intervention protocols [37]. To date, no studies have specifically examined the effects of VD supplementation on HE, though some have looked at its effects indirectly [39,42].

Additionally, research should investigate the role of VD in enhancing outcomes for patients undergoing TIPS procedures, potentially improving clinical management and reducing complications. Given VD’s association with the amelioration of oxidative stress, examining serum VD levels in patients undergoing TIPS for treatment of PH may yield promising results. Studies like that of Jha et al. can also serve as a foundation for designing future trials that specifically target HE and explore whether VD supplementation could mitigate cognitive and neuromuscular impairments associated with this condition [37]. The relationship between VDD and adverse clinical outcomes warrants further exploration, particularly through long-term studies tracking health outcomes in deficient individuals. The interaction between VDD, HE, and type 2 diabetes mellitus (T2DM) also represents a critical area for investigation. A focus on whether VD supplementation can mitigate cognitive impairments and improve metabolic control in these patients is essential.

The antioxidant properties of VD and its role in suppressing oxidative stress should be examined in various pathological conditions characterized by elevated oxidative stress, such as CVD and neurodegenerative disease. The downstream effects of oxidative stress may be the underlying mechanism of disability in other disease states like HE and T2DM. Therefore, further understanding of VD’s detoxifying role may uncover potential therapeutic advantages in these diseased states. Advanced mechanistic studies and clinical trials are essential to uncover the full therapeutic potential of VD and to establish evidence-based guidelines for its supplementation. These studies should also evaluate variations in dosing strategies, such as high dose loading protocols followed by maintenance therapy, to determine the most effective regimens for different populations and disease states. Such research could potentially enhance our understanding of VD’s complex roles and inform clinical practices on optimizing health outcomes across a range of conditions.

## 7. Conclusions

HE is a prevalent neuropsychiatric disorder associated with liver cirrhosis and PSS procedures. VDD is common among cirrhotic patients, and its role in HE management has gained increasing attention. Current evidence suggests a significant correlation between VDD and HE severity, with lower VD levels associated with worse symptoms. The pathophysiology of cirrhosis involves hepatocyte death, fibrosis, and oxidative stress, all contributing to HE. Ammonia buildup, a key contributor to HE, leads to oxidative imbalance and neuroinflammation. VD, with its antioxidant capabilities, may help mitigate this oxidative stress, presenting a potential therapeutic advantage. Future research should prioritize longitudinal studies to assess the impact of VD on mortality, HE incidence, and severity, particularly in high-risk patients undergoing TIPS procedures. Additionally, exploring VD’s antioxidant properties across conditions characterized by oxidative stress could reveal broader therapeutic potential.

## Figures and Tables

**Figure 1 nutrients-16-04007-f001:**
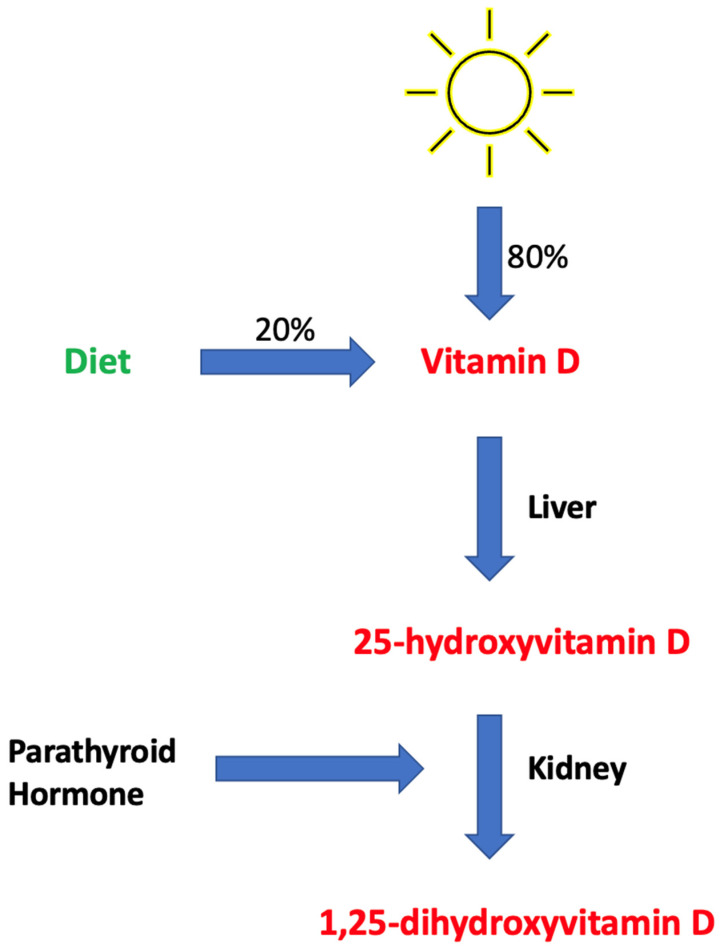
The biological process by which vitamin D is hydroxylated twice and converted into 1,25-dihydroxyvitamin D.

**Table 1 nutrients-16-04007-t001:** Studies published after 2010 showing an association between vitamin D deficiency and hepatic encephalopathy.

References	Title	Disease (*n*)	Prevalence of VDD	Finding(s)
Putz-Bankuti et al., 2012 [34]	Association of 25-hydroxyvitamin D levels with liver dysfunction and mortality in chronic liver disease	CLD (*n* = 75)	VDD (<20 ng/mL) in 53(71%) of patients	25-OHD levels were inversely correlated with CP and MELD scores
Savic et al., 2014 [40]	Vitamin D status, bone metabolism and bone mass in patients with alcoholic liver cirrhosis	ALD (*n* = 30)	VDD (<50 nmol/L) in 67% of patients	VDD had the highest prevalence in CP C class patients
Vidot et al., 2017 [41]	Serum 25-hydroxyvitamin D deficiency and hepatic encephalopathy in chronic liver disease	CLD (*n* = 165)	Moderate to severe 25-OHD deficiency was identified in 49 patients of whom 36 had grade 2–3 HE	25-OHD deficiency is observed in many patients with CLD
Jha et al., 2017 [42]	Effect of replenishment of vitamin D on survival in patients with decompensated liver cirrhosis: A prospective study	Decompensated CLD (*n* = 153)	VDD (<20 ng/mL) in 129(84%) of patients	VD supplementation may increase survival probability of patients with decompensated liver cirrhosis with improvements in MELD and CP scores
Khan et al., 2019 [43]	Impact of Vitamin D Status in Chronic Liver Disease	CLD patients (*n* = 75)	VDD (<20 ng/dL) was found in 41% out of which 19% suffered from severe VDD (<10 ng/dL)	VDD was associated with CLD and an increased in CP score
Yousif et al., 2019 [44]	Associated vitamin D deficiency is a risk factor for the complication of HCV-related liver cirrhosis including hepatic encephalopathy and spontaneous bacterial peritonitis	HCV-related liver cirrhosis w/and w/o HE (*n* = 90)	25-OHD level was on average 16.28 ng/mL in control and 6.81 ng/mL in HE group	Low serum levels of 25-OHD were associated with HE in cirrhotic patients
Narayanasamy et al., 2019 [45]	High Prevalent Hypovitaminosis D Is Associated with Dysregulation of Calcium-parathyroid Hormone-vitamin D Axis in Patients with Chronic Liver Diseases	CLD patients (*n* = 236)	VDD (<30 ng/dL) was found in 162 (69%)	VDD is associated with higher CP scores
Kumar et al., 2020 [38]	Serum 25-hydroxyvitamin D level in patients with chronic liver disease and its correlation with hepatic encephalopathy: A cross-sectional study	CLD w/and w/o HE (*n* = 100)	Severe 25-OHD deficiency was seen in 38% of HE group compared to 6% in control group	Serum 25-OHD deficiency was more prevalent in HE group compared to control
Simbrunner et al., 2020 [46]	Vitamin A levels reflect disease severity and portal hypertension in patients with cirrhosis	ACLD (*n* = 234)	VDD was found in 133 (57%)	VDD increased with increasing CP stages
Afifi et al., 2021 [6]	Low Serum 25-Hydroxy Vitamin D (25-OHD) and Hepatic Encephalopathy in HCV-Related Liver Cirrhosis	HCV-related liver cirrhosis w/and w/o HE (*n* = 100)	HE groups w/severe VDD was 16% compared to other group at 6%; HE group w/moderate VDD was 24% compared to other group at 10%	Lower levels of 25-OHD were associated with higher incidence of HE in cirrhotic HCV patients
Kalita et al., 2022 [7]	Vitamin D in Patients of Chronic Liver Disease with Hepatic Encephalopathy	CLD w/HE (*n* = 88)	Mean serum 25-OHD was 24.11, 13.61, 8.41, and 8.00 ng/mL in grades 1–4 HE, respectively	Mean serum 25-OHD deficiency became more severe as HE worsened
Sarkar et al., 2024 [5]	Association of low serum 25-Hydroxy vitamin D [25(OH)D] with hepatic encephalopathy in patients with decompensated liver cirrhosis	Decompensated cirrhosis of the liver w/and w/o HE (*n* = 70)	91% of HE patients had moderate to severe 25-OHD deficiency compared 51% in control group	Significant association was found between low serum 25-OHD and HE

Abbreviations: HE, hepatic encephalopathy; HCV, Hepatitis C virus; VDD, vitamin D deficiency; 25-OHD, 25-Hydroxy vitamin D; CLD, chronic liver disease; CP, Child–Pugh; ACLD, advanced chronic liver disease; ALD, alcoholic liver disease; MELD, model for end-stage liver disease; VD, vitamin D.

## Data Availability

The figure and table used in the current article are available from the corresponding author on reasonable request.

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
