# Peer review of "The Role of Vitamin D Deficiency in Hepatic Encephalopathy: A Review of Pathophysiology, Clinical Outcomes, and Therapeutic Potential"

_nutrients, 2024, doi:10.3390/nu16234007_

Round 1

Reviewer 1 Report

Comments and Suggestions for Authors

In this review, the authors investigated the association between Vitamin D deficiency and hepatic encephalopathy. This review is well-written and addresses an important topic. However, the authors must provide more detail on a few topics.

- Topic 2: The authors could further explore the clinical characteristics (neurocognitive and neuromuscular manifestations).

- It would be important for the authors to present the doses used in the studies that used vitamin D supplementation. For example, in Table 1, what dose was used by Jha et al. (2017) [reference number 37]? This was little discussed by the authors in topic 6 (lines 268-270). As the authors presented “What Should Future Studies Explore?”, a more detailed discussion of the possible intervention characteristics (type, dose, duration, etc.) of vitamin D supplementation could enrich the review.

- Line 41 - add the full name of VDD before the acronym, as done for the others.

Author Response

Comments 1: Topic 2: “The authors could further explore the clinical characteristics (neurocognitive and neuromuscular manifestations).”

Response 1:

Thank you for highlighting this important point. We have expanded the description of the neurocognitive and neuromuscular manifestations of hepatic encephalopathy (HE) to provide greater detail. Specifically, we elaborated on the neurological features, such as attention deficits, slowed psychomotor speed, and executive dysfunction, as well as neuromuscular features like tremors, asterixis, and extrapyramidal symptoms. These revisions can be found on page 2, section 2, lines 53-74.

Updated Text in Manuscript:

Patients may present with subtle neuropsychological impairments, such as attention deficits and slowed psychomotor speed, progressing to more overt symptoms like confusion, somnolence, and disorientation. In severe cases, HE may progress to stupor or coma. Psychometric testing often reveals deficits in visuospatial ability, working memory, and executive function, which may not be evident on routine neurological examination [8, 10].

Neuromuscular manifestations of HE is also significant and may include tremors, rigidity, and reduced motor coordination. Asterixis, characterized by sudden, brief, nonrhythmic lapses in posture, is a hallmark sign of HE. In advanced stages, extrapyramidal symptoms such as bradykinesia and hypertonia may develop, with some patients showing signs of corticospinal tract involvement, such as spasticity or hyperreflexia [8]. These symptoms are often accompanied by alterations in the sleep-wake cycle and motor slowing, which can significantly impact daily functioning [9, 10].

The pathophysiology underlying these manifestations is complex, involving a combination of hyperammonemia, neuroinflammation, and astrocyte swelling. Hyperammonemia, resulting from impaired ammonia metabolism due to liver dysfunction or portosystemic shunting, leads to the accumulation of glutamine within astrocytes, contributing to cerebral edema and neurotransmitter imbalances [9, 10]. This process disrupts synaptic transmission and contributes to both cognitive and motor impairments. Neuroinflammation, characterized by microglial activation and increased proinflammatory cytokines, further exacerbates these effects, leading to increased glutamatergic and GABAergic signaling that contributes to the characteristic neurological symptoms of HE [10].

Comments 2: “It would be important for the authors to present the doses used in the studies that used vitamin D supplementation. For example, in Table 1, what dose was used by Jha et al. (2017) [reference number 37]? This was little discussed by the authors in topic 6 (lines 268-270). As the authors presented “What Should Future Studies Explore?”, a more detailed discussion of the possible intervention characteristics (type, dose, duration, etc.) of vitamin D supplementation could enrich the review.”

Response 2:

Thank you for this suggestion. Only one study in the table used VD supplementation, so the authors did not want to include the dosing in the table since it was just one study; however, we have addressed this concern by incorporating the dosing details from Jha et al. (2017) into the discussion sections of the manuscript. Specifically, we added information about the supplementation protocol described in this study in Section 4.4: Vitamin D Levels in Association with Hepatic Encephalopathy and Section 6: What Should Future Studies Explore?

The revised text in Section 4.4 explicitly describes the dosing regimen, including a loading dose of 300,000 IU of intramuscular cholecalciferol followed by 800 IU/day oral maintenance and 1,000 mg of calcium daily, along with its association with improvements in CP scores and cirrhosis-related outcomes. In Section 6, this information is used to highlight the potential for such protocols to inform future clinical trials on HE.

Updated Text in Manuscript:

Page 6, section 4.4, lines 224-228:
Patients with elevated CP scores showed a decline in scores over a 6-month period while receiving vitamin D supplementation, which included a loading dose of 300,000 IU intramuscular cholecalciferol followed by 800 IU/day oral maintenance, along with 1,000 mg of calcium daily. This regimen indicated an overall positive effect of VD supplementation on cirrhosis mortality [37].

Page 8-9, section 6, lines 309-316, 322-325, 338-341, respectively:
For example, Jha et al. (2017) demonstrated that a regimen of vitamin D supplementation—comprising a single loading dose of 300,000 IU intramuscular cholecalciferol followed by daily oral maintenance of 800 IU, alongside 1,000 mg of calcium—yielded promising results in improving clinical outcomes in patients with decompensated cirrhosis. However, the study’s findings on survival rates did not reach statistical significance, highlighting the need for larger, multicenter trials to confirm these effects and optimize intervention protocols [37].

Studies like those of Jha et al. can also serve as a foundation for designing future trials that specifically target, HE and explore whether VD supplementation could mitigate cognitive and neuromuscular impairments associated with this condition [37].

These studies should also evaluate variations in dosing strategies, such as high dose loading protocols followed by maintenance therapy, to determine the most effective regimens for different populations and disease states.

Comment 3: “Line 41 - add the full name of VDD before the acronym, as done for the others.”

Response 3:

We agree with this suggestion and have clarified the full name as “Vitamin D Deficiency (VDD)” before using the acronym on page 1, section, 1, line 42-43.

Reviewer 2 Report

Comments and Suggestions for Authors

This is a notable review of the complexities linking vitamin D deficiency to hepatic cirrhosis and encephalopathy. A range of disorders is reviewed with various explanations discussed as to the mechanism by which hepatic disease is associated with vitamin D deficiency and consequent neurological dysfunction. It would help the analysis if the work of Clements et al. (doi: 10.1111/j.1365-2265.1992.tb02278.x.) on the development of vitamin D deficiency in hyperparathyroidism were also considered. The work of that group indicates how hepatic inactivation of 25-hydroxyvitamin D when production of 1,25-dihydroxyvitamin D is stimulated by parathyroid hormone leads to vitamin D deficiency. The subsequent studies of Gressner et al. (doi: 10.1016/j.cca.2007.12.011.)  showed that vitamin D deficiency leads to uptake of the vitamin D-binding protein by hepatic stellate cells which then are transformed into myofibroblasts, as a component of the pathological development of cirrhosis. Hence these hepatic changes associated with vitamin D deficiency could well also be considered as factors in the link between hepatic damage and subsequent encephalopathy.

Author Response

Comments 1: It would help the analysis if the work of Clements et al. (doi: 10.1111/j.1365-2265.1992.tb02278.x.) on the development of vitamin D deficiency in hyperparathyroidism were also considered. The work of that group indicates how hepatic inactivation of 25-hydroxyvitamin D when production of 1,25-dihydroxyvitamin D is stimulated by parathyroid hormone leads to vitamin D deficiency.

Response 1: Thank you for pointing this out. We agree with this comment. Therefore, we have added this information on page 5 section 4.3 lines 189-193. It reads: Additionally, there is evidence to suggest that hyperparathyroidism in cirrhotic patients may exacerbate VDD.  Clements et al. concluded that when production of 1,25-dihydroxyvitamin D is stimulated by parathyroid hormone, this further exacerbates VDD via hepatic inactivation of 25(OH)D [32]. This process underscores the interplay between hormonal regulation and hepatic function in VD metabolism.

Comments 2: The subsequent studies of Gressner et al. (doi: 10.1016/j.cca.2007.12.011.)  showed that vitamin D deficiency leads to uptake of the vitamin D-binding protein by hepatic stellate cells which then are transformed into myofibroblasts, as a component of the pathological development of cirrhosis. Hence these hepatic changes associated with vitamin D deficiency could well also be considered as factors in the link between hepatic damage and subsequent encephalopathy.

Response 2: Again, thank you for pointing this out. We agree with this comment. Therefore, we added this information on page 3, section 3, lines 103-107. It reads, “Additionally, VDD leads to uptake of vitamin D-binding protein by hepatic stellate cells. This triggers their transformation into myofibroblasts resulting in the development of hepatic cirrhosis. These pathological changes not only contribute to cirrhosis but may also play a role in the progression of HE by altering hepatic architecture and function.”